# Monitoring, Delivery and Outcome in Early Onset Fetal Growth Restriction

**Andrea Dall'Asta** (ID)**, Monica Minopoli** (ID)**, Tullio Ghi and Tiziana Frusca ***

Department of Medicine and Surgery, Obstetrics & Gynecology Unit, University of Parma, 43121 Parma, Italy; andrea.dallasta1@gmail.com (A.D.); minopoli.monica@gmail.com (M.M.); tullio.ghi@unipr.it (T.G.)
* Correspondence: fruscatiziana@gmail.com or tiziana.frusca@unipr.it; Tel.: +39-0521702434

**Abstract:** Early fetal growth restriction (FGR) remains a challenging entity associated with an increased risk of perinatal morbidity and mortality as well as maternal complications. Significant variations in clinical practice have historically characterized the management of early FGR fetuses. Nevertheless, insights into diagnosis and management options have more recently emerged. The aim of this review is to summarize the available evidence on monitoring, delivery and outcome in early-onset FGR.

**Keywords:** maternal–fetal Doppler; perinatal complications; antenatal monitoring; preterm delivery; neonatal intensive care unit





## 1. Introduction

Fetal growth restriction (FGR) complicates approximately 10% of all pregnancies and is among the leading causes of perinatal morbidity and mortality [1–3].

Fetal growth restriction is conventionally differentiated in early or late FGR. These two entities differ not only on the basis of the gestational age at diagnosis, which has been conventionally established at 32 weeks, but also in terms of clinical features, severity of placental dysfunction and maternal morbidity [4–6]. Severe placental dysfunction and up to 70% association with hypertensive disorders of the pregnancy (HDP) are among the features characterizing early FGR, which accounts for approximately 20–30% of all cases of FGR [4,7].

FGR is a complex and multifactorial disorder affecting fetal development. Most cases are related to uteroplacental dysfunction [8], while non-placental etiologies include chromosomal/genetic anomalies, congenital infections [9] and inborn errors of metabolism [10].

FGR often results in multiple perinatal complications [11–13] and is an acknowledged risk factor for poor neurological outcome and cardiovascular disease, including hypertension, diabetes and dyslipidemia, in children and adults [14,15]. The risk of adverse perinatal and long-term outcome is directly related to the severity of the growth retardation and the gestational age at delivery.

At present, no treatment can reverse the course of FGR. Two recent clinical trials conducted with the aim of investigating a potential role of sildenafil citrate in improving fetal growth in utero could not demonstrate any clinical benefit compared to the administration of a placebo [16,17], and in one such study [17], an increased frequency of pulmonary hypertension was reported in FGR fetuses antenatally exposed to sildenafil citrate.

On this basis, to date, the only treatment option is represented by timed delivery, i.e., required when the risk of intrauterine compromise outweighs that of prematurity. However, there still is a great variation in clinical practice when it comes to FGR monitoring and timing of delivery.

The aim of this review is to summarize current evidence regarding monitoring and timing of delivery in structurally normal fetuses diagnosed with early FGR of suspected uteroplacental cause.

## 2. Definition and Diagnosis

Multiple definitions of FGR have been suggested over the decades by national and international societies [4,7,18–23].

The Delphi Consensus criteria proposed by a panel of European Fetal Medicine experts for the definition of early FGR include either severe fetal smallness or umbilical artery (UA) Doppler late abnormalities alone, or a combination of fetal smallness and milder abnormalities of the UA Doppler or uterine artery (UtA) Doppler detected before 32 weeks [24]. Such a definition summarizes the current understanding of the pathogenesis of non-anomalous FGR, which consists of pathological smallness caused by an underlying functional problem. Such a definition of early FGR, including biometric cut-off and Doppler indices of feto-placental function, is currently endorsed by most fetal medicine specialists [4,7,24–26]. Accurate dating in the first trimester of the pregnancy is essential for the correct diagnosis of FGR [27,28].

The evaluation of the fetal anatomy and of the amniotic fluid volume can assist in the differential diagnosis of the underlying etiology of the fetal smallness (e.g., uteroplacental cause, viral infection, karyotype abnormality or genetic syndromes). The role of invasive testing for fetal karyotyping has been revised following the results of one study by Borrel showing 6.8% higher detection of genetic defects by means of genomic microarray compared to karyotyping in fetuses with FGR diagnosed prior to 32 weeks [29]. On this basis, the recent SMFM guidelines [30] recommend invasive testing in the event of FGR diagnosed before 32 weeks.

## 3. Monitoring Tools in Early-Onset Fetal Growth Restriction

Early-onset FGR of utero-placental origin epitomizes a clinical phenotype characterized by increased resistance of the utero-placental circulation, which in turn leads abnormally elevated umbilical artery blood flow resistance [31]. Differently from late-onset FGR, early FGR usually shows a pattern of Doppler deterioration which is related to the degree of the placental dysfunction, starting from abnormalities in the UA Doppler and then involving sequentially the middle cerebral artery (MCA) and the ductus venosus (DV) [32]. Such a temporal sequence of longitudinal changes in fetal circulation in early FGR fetuses was first proposed by Ferrazzi et al. In detail, absent end-diastolic flow (AEDF) in the umbilical artery and vasodilatation in the middle cerebral arteries were identified as early changes, while reversed end-diastolic flow (REDF) and abnormalities in the DV Doppler were depicted as late changes associated with an increased risk of adverse perinatal events [3].

Routine evaluation of the UA Doppler has been proven to reduce perinatal morbidity and mortality [22], being able to provide both diagnostic and prognostic information for the management of FGR [33]. For this reason, the Royal College of Obstetrician and Gynecologists recommends its use as a primary surveillance tool in fetuses with confirmed or suspected FGR [22]. Furthermore, as stated above, abnormalities of the UA Doppler are among the currently acknowledged criteria for the confirmation of pathological smallness [4,24]. Different degrees of impaired placental function can be identified by means of the assessment of the pulsatility index (PI) and the UA end-diastolic flow (EDF). For example, absent end-diastolic flow (AEDF) or reversed end-diastolic flow (REDF) indicate an important reduction in placental function. Former studies on high-risk pregnancies have shown that the transition from AEDF to REDF may be slow and gradual in early FGR and last for days or weeks before the appearance of abnormal heart rate patterns which indicate the need for immediate delivery [34]. Nonetheless, such abnormal Doppler patterns have been associated with a significant risk of perinatal morbidity and mortality [35] and also with a higher incidence of long-term permanent neurologic damage when compared to FGR fetuses with positive UA EDF [36].

A decrease in MCA resistance, which usually follows the changes in the UA Doppler, can be identified by means of a reduction in the MCA PI. The so-called brain sparing effect, which consists of vasodilatation of the brain circulation, represents an adaptive

mechanism to chronic hypoxia secondary to uteroplacental insufficiency. Of note, abnormal Doppler patterns in the UA and MCA have been related to histological signs of placental insufficiency [37]. Available data suggest an association between cerebral vasodilatation and adverse perinatal and neurological outcome in FGR fetuses, regardless of umbilical artery Doppler [7,33,38,39], even though available evidence does not support the routine use of the MCA Doppler or its ratios (cerebroplacental ratio, CPR, or umbilico-cerebral ratio, U-C ratio) for the diagnosis and management of early FGR.

The Doppler examination of the ductus venosus (DV) plays a crucial role in the management of early FGR according to the protocol described by the Trial of Randomized Umbilical and Fetal Flow in Europe (TRUFFLE) group [4,40,41]. The DV is responsible for the shunting of the oxygenated blood from the umbilical vein to the right atrium during fetal life. In normal conditions approximately 15–30% of the umbilical blood is shunted, while this percentage increases in FGR fetuses in order to improve cerebral and cardiac perfusion. This mechanism is dependent on the vasodilatation of the DV, which is mediated by nitric oxide and prostaglandin released in response to fetal hypoxia, and on the increasing impedance in the umbilical artery [42]. The waveform sampled from the DV consists of a biphasic wave showing an "S" component, which corresponds to the ventricular systole, a "D" component, which corresponds to the early ventricular diastole, and an "A" component, which corresponds to the late ventricular filling, which is dependent on the atrial contraction. Abnormal DV waveforms show a gradual increase in the PI, which is then followed by an absent and inverted "A" wave as a result of the increasing ventricular preload associated with the vasodilatation of the isthmus of the DV [43]. In the randomized trial conducted by the TRUFFLE group, "early" changes in the Doppler pattern measured in the DV were defined by a pulsatility index above the 95th centile, while "late" changes were defined by an absent or inverted "A" wave [4,40].

Computerized cardiotocography (cCTG) is the only tool which allows the quantitative analysis of the STV of the fetal heart rate, which represents an important indicator of fetal wellbeing [44]. cCTG is acknowledged to play a major role in the management of early FGR [4,40,45]. In such cases, STV < 2.6 milliseconds has been related to fetal acidemia and intrauterine death [40,44], while STV > 3.0 milliseconds has been rarely associated with poor fetal outcome [44,45]. With regard to the available knowledge on the longitudinal changes in early FGR, while UA and MCA Doppler abnormalities occur in the early phase of the fetal deterioration, the short-term variation (STV) in the fetal heart rate, similarly to DV flow waveforms, becomes abnormal in the advanced stages of fetal compromise [46–49]. In 2001, Hecher et al. [44] evaluated the longitudinal trend of the parameters used for monitoring in FGR. In a cohort of 110 fetuses defined as early FGR based on abdominal circumference below the fifth centile between 24 and 34 weeks, the authors demonstrated that DV PI and STV show mirroring trends, with the DV PI increasing and the STV decreasing, starting from 21 days before the need to expedite delivery due to fetal distress. Interestingly, the regression lines crossed the respective limits of +2 SD and –2 SD at almost the same point in time, namely approximately 7 days before delivery. Furthermore, both parameters showed good correlation with perinatal outcome. On this basis, the authors concluded that the DV PI and the cCTG STV represent crucial parameters to be evaluated when deciding the timing of delivery before 32 weeks of gestation [44].

Additionally, Doppler abnormalities which appear in stages of advanced fetal compromise, such as REDF in the umbilical artery and reversed "A" wave in the ductus venosus, are associated with a significant risk of perinatal mortality [3]. It has been estimated that abnormal DV precedes the loss of short-term variability in computerized cardiotocography (cCTG) in around 50% of cases [49] and precedes abnormalities in the biophysical profile by 48–72 h. Of note, the PI of the DV has been inversely related to the cord pH at birth in FGR fetuses, and absent or reverse velocities in the DV during atrial contraction have been associated with perinatal mortality independently of the gestational age at delivery [49].

## 4. Management and Delivery in Early-Onset Fetal Growth Restriction: What We Have Learned from the TRUFFLE Study

At present, the TRUFFLE is the only randomized controlled study which has evaluated a standardized monitoring and delivery protocol for FGR fetuses and has demonstrated its effectiveness in optimizing the short- and long-term outcome of non-anomalous FGR fetuses diagnosed between 26 and 32 weeks.

Based on the assumption that a monitoring strategy including the cCTG STV and the DV Doppler can allow practitioners to safely delay delivery before the occurrence of fetal compromise, the TRUFFLE study has demonstrated that the perinatal outcome of surviving early FGR fetuses is significantly better among those delivered based on late DV changes [4,40,45,50], even though no differences were noted when evaluating the primary outcome of the study, i.e., survival without neurodevelopmental impairment among the three randomization arms of the TRUFFLE. According to the results of the study, the DV Doppler represents the most important parameter in the prediction of intrauterine death in early-onset FGR [33]. With regard to the perinatal outcomes, an absent or reversed DV "A" wave has been associated with late-stage acidemia and a 40–70% risk of fetal death irrespective of gestational age and has been shown to shortly precede the onset of spontaneous deceleration on CTG monitoring [33]

The cCTG STV was reported to have a similar performance as that of reversed or absent DV "A" wave in the prediction of fetal death, and previous data also demonstrated that abnormalities of the cCTG STV occur in the event of advanced fetal deterioration [47]. Although the optimal STV cut-off value for delivery has yet to be clarified, it is important to point out that between 26 and 32 weeks, expectant management is accepted as long as either the DV or the STV is abnormal but not if both are abnormal [4].

Another randomized trial previously demonstrated the benefits of expectant management in FGR fetuses; however, the inclusion criteria in the GRIT were not as strict as those of the TRUFFLE study, and the decision on how to monitor and when to deliver FGR fetuses was not standardized [51].

The estimated fetal weight (EFW) and gestational age represent crucial factors to be considered when managing and counseling in early FGR [50,52]. Indeed, EFW and gestational age thresholds for fetal viability need to be considered both when evaluating the options of termination of pregnancy—when legally admitted—and during invasive testing and delivery of a potentially viable fetus.

"Safety net" criteria for delivery within the TRUFFLE cohort included spontaneous decelerations at CTG, UA REDF between 30 and 32 weeks, UA AEDF between 32 and 34 weeks or UA PI > 95th centile beyond 34 weeks. Therefore, according to the TRUFFLE protocol, the abnormalities of the UA Doppler should not be considered when evaluating the option of delivery prior to 30 weeks of gestation [4,40].

Importantly, the "safety net" criteria accounted for a significant number of indications for delivery, both in the primary [4,40] and in a recently published secondary analysis of the datasets including only the cases delivered < 32 weeks, mostly within the late DV group [45].

The umbilical artery Doppler becomes the most important parameter to assess the timing of delivery beyond 32 weeks of gestation. More specifically, according to the TRUFFLE protocol, delivery is recommended between 32 and 34 weeks of gestation in the case of AEDF occurrence, while, beyond 34 weeks, delivery should be considered in the event of UA PI > 95th centile [4,40]. Such recommendations are not consistent with those of the recent guidelines on diagnosis and management of FGR published by the International Society of Ultrasound in Obstetrics and Gynecology, which recommend delivery in the event of UA AEDF beyond 34 weeks and UA PI above the 95th percentile beyond 36 weeks [53].

With respect to cerebral redistribution, which represents an adaptive mechanism to fetal hypoxemia and can be identified by means of a reduction in the impedance in the middle cerebral artery (MCA) and in the cerebroplacental ratio (CPR) (or cerebro-umbilical

(C-U) ratio), there is no evidence supporting its role in the monitoring strategy of early FGR fetuses. Even though an association between cerebral vasodilatation and an increased risk of brain damage leading to subsequent neurodevelopmental impairment in FGR fetuses has also been suggested [39,54], there is a paucity of prospective, good-quality studies with adequate sample size and long-term follow-up which have addressed this issue. Anticipated delivery of the fetuses showing signs of cerebral redistribution has not been demonstrated to add any benefit on short- or long-term outcomes [33,54,55], and the TRUFFLE protocol does not endorse the MCA Doppler or CPR (or C-U ratio) for the management of early FGR fetuses. According to a secondary analysis of the study, the MCA PI changes over time and measured close to delivery showed no impact on neonatal and 2-year neurodevelopmental outcome, which led to the conclusion that gestational age at delivery remains the most important factor in determining neonatal survival without adverse outcome and, together with birthweight, infant outcome [56].

A recently published secondary analysis of the TRUFFLE cohort dataset regarding the longitudinal changes in the STV has shown that it is not possible to predict the occurrence of abnormal STV or late changes in the DV Doppler, thus concluding that, in the case of advanced fetal compromise, cCTG monitoring should be undertaken at least on a daily basis [57], while monitoring of fetal Doppler can be performed twice a week or on alternate days in the event of advanced fetal compromise.

Biophysical profile and conventional CTG, as well as uterine artery Doppler, have no role in the TRUFFLE protocol for the monitoring of severely growth-restricted fetuses given the lack of data supporting their usefulness in the management of early FGR [58–61].

Furthermore, there are no data concerning whether to recommend a strategy of inpatient versus outpatient monitoring of early FGR fetuses. While most cases of isolated FGR are monitored in an outpatient setting, we believe that the decision for inpatient monitoring should be considered on an individual basis. Of note, 60–70% of cases of early FGR are associated with hypertensive complications of the pregnancy [4]. In such cases, particularly in the case of PE, admission seems advisable despite the lack of clinical data supporting the approach.

There is still no evidence in the literature regarding the monitoring of early FGR. We present the management protocol currently adopted in our clinical practice in Table 1.

**Table 1.** Proposed management of non-anomalous FGR prior to 32 weeks.

| 22+0–25+6 | | 26+0–31+6 | |
|---|---|---|---|
| EFW < 500 g, AREDF in UA | Follow-up at 26+0 weeks | EFW <10°P and PI in UA >95°P with normal DV | Twice weekly follow-up |
| EFW > 500 g | Personalized management | EFW <3°P and normal UA Doppler | Weekly follow-up |
| | | EFW 3°P-10°P normal UA Doppler | Fortnightly follow-up |
| | | AREDF ➜ consider admission | |
| | | Daily cCTG | |
| | | Doppler every 2–3 days | |
| | | 26+0–28+6 | 29+0–31+6 |
| * If abnormal findings are confirmed in two measurements, delivery indicated after steroid administration and Magnesium Sulphate | | DV a-wave at or below baseline and/or STV < 2.6 and/or spontaneous decelerations * | DV a-wave at or below baseline and/or STV < 3.0 and/or spontaneous decelerations and/or REDF >30+0 weeks * |

FGR = fetal growth restriction; EFW = estimated fetal weight; AREDF = absent or reversed end diastolic flow; UA = um-bilical artery; PI = pulsatility index; DV = ductus venosus; STV = short-term variability; cCTG = computerized cadioto-cography.

## 5. Early-Onset Fetal Growth Restriction in the TRUFFLE Era: Delivery and Fetal Outcomes

The identification of the optimal timing of delivery represents the crucial clinical challenge in the management of early FGR fetuses, as it requires a balance between the risks of prematurity and stillbirth and those of severe intrauterine hypoxia with organ damage due to inadequate tissue perfusion [33,62]. The TRUFFLE group has designed a reliable protocol for the monitoring and the identification of the optimal timing of delivery in early FGR, unless severe maternal complications supervene [4,51,63]. According to the data from the earlier papers from the group, overall survival and survival without neurodevelopmental morbidity showed remarkably higher than expected percentages [4,40], 92% and 70%, respectively. Intrauterine deaths accounted for only 2% of the included cases, while cerebral palsy was reported in only six fetuses (1%) within this cohort of preterm and severely growth-restricted fetuses.

Such findings were subsequently confirmed in a secondary analysis of the TRUFFLE datasets which focused on all fetuses delivered before 32 weeks, who were delivered based on the TRUFFLE protocol [45]. However, it is important to note that only 11/66 (16.7%) of these fetuses were delivered based on their allocation to the "late DV" group, while the majority were delivered due to either fetal safety nets or other indication including maternal morbidity.

As regards the mode of delivery, there is no recommendation by the TRUFFLE group as to whether to deliver vaginally or by cesarean section, although 97% of the included women underwent cesarean delivery. Such a percentage is similar to that formerly reported by Baschat et al. [18] and higher than that of the GRIT group [51]. Of note, the recent ISUOG guidelines on the diagnosis and management of FGR recommend elective cesarean delivery in the presence of any among abnormal cCTG STV, ductus venosus Doppler alteration, absent or reversed UA-EDF, altered blood pressure or maternal indication [53].

## 6. Periviable Fetal Growth Restriction

While the perinatal and the 2-year neurodevelopmental outcomes of FGR diagnosed between 26 and 32 weeks has been described in the TRUFFLE randomized trial [4,40,45,64,65], little evidence exists for counseling the prospective parents when a diagnosis of FGR is made at periviable gestation. Pregnancies with very small fetuses near the limit of viability remain a challenge for the clinician in terms of counseling and management. The recently published ISUOG guidelines [53] recommend personalized management up to 26 weeks of gestation; therefore, active management in terms of monitoring and delivery can be deferred until 26 weeks is reached in order to improve the chance of survival and disease-free survival, particularly in the context of fetuses who have not reached a "viable" EFW.

With regard to non-anomalous FGR, three retrospective studies have reported different results in terms of perinatal survival, which was explained on the basis of the different criteria adopted for the definition of FGR [8,66,67]. While the study by Temming et al. described obstetric and neonatal outcomes of fetuses diagnosed as FGR based on an EFW < 10th percentile between 17 and 22 weeks [67], another retrospective study by Lawin-O'Brien et al. [8] reported the perinatal outcomes of 245 fetuses defined as FGR based on an AC CA below the third centile recorded between 22 and 26 weeks of gestation, showing poorer outcomes compared to the study by Temming et al. [67] in terms of mean gestational age at delivery (27.7 vs. 37.2 weeks), mean birthweight (1020 vs. 2725 g) and incidence of stillbirth (36% vs. 2.5%) and neonatal death (9% vs 1.4%). A third case-series of pregnancies complicated by FGR diagnosed between 18 and 25 weeks of gestation and defined by an EFW below the third percentile aligns with the latter, reporting an incidence of stillbirth and neonatal death of 30% and 10%, respectively [66].

With respect to FGR fetuses associated with structural defects or chromosomal abnormalities, to our knowledge, only one case-series has reported the short-term outcomes of 52 anomalous FGR fetuses diagnosed between 22 and 26 weeks of gestation. Within

the limitations of the small case-series, the reported perinatal survival was not dissimilar from that of non-anomalous FGR paired for gestational age at diagnosis, even though the diagnosis of a genetic abnormality associated with the fetal smallness proved to be invariably lethal [68]. Of note, in this single-center case-series, the anomalous FGR fetuses accounted for almost one third of all the fetuses identified as FGR at periviable gestation, thus highlighting the importance of a thorough assessment of the fetal anatomy when fetal smallness is diagnosed prior to 26 weeks.

## 7. Conclusions

This review summarizes the current knowledge on FGR diagnosed prior to 32 weeks of gestation. A standardized protocol integrating Doppler and cCTG parameters for the monitoring of the pregnancies complicated by early FGR has been developed, and available evidence supports its use for the management of FGR between 26 and 32 weeks in order to optimize the perinatal outcome as well as the survival without neurodevelopmental delay of preterm FGR fetuses. Delivery should be undertaken only if either the DV or the STV become abnormal, and available evidence suggests that, once the placental origin of the growth restriction is confirmed, the perinatal and infant outcomes are better than formerly reported.

**Funding:** This research received no external funding.

**Institutional Review Board Statement:** Not applicable.

**Informed Consent Statement:** Not applicable.

**Conflicts of Interest:** The authors declare no conflict of interest.

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
