# Peer review of "Monitoring, Delivery and Outcome in Early Onset Fetal Growth Restriction"

_2673-3897, doi:10.3390/reprodmed2020009_

Round 1

Reviewer 1 Report

Very nice overview of the management of early FGR fetuses. Good article structure and overview of the evidence. 

Since this topic is a very intriguing and actual field of perinatology with many questions and contradictions I would recommend that authors summarize their recommendations in a form of management algorithm (for example figure 1.) according to the gestational weeks (x axis). 

Otherwise I recommend publishing the article. 

Author Response

Dear reviewer,

thank you for your suggestion. We added a table with a proposition of FGR management aligned with our clinical practice.

You can find it in attachment

Reviewer 2 Report

This review well summarized current evidence regarding monitoring and timing of delivery in structurally normal fetuses diagnosed with early FGR.

Author Response

Thank you for the favourable comment.

Reviewer 3 Report

The aim of this manuscript  is to summarize the available evidence on monitoring, delivering and outcome in early onset FGR. The authors have summarized the most important works in this field, as well as the most important cornerstones. The manuscript is written well and Englisch is used correct and readable.

Although, no new knowledge has been introduced this paper provides a brief and pregnant summary of the management of early fetal growth retardation and is therefore suitable for publication

Author Response

Thank you for the comment.
Indeed this is just an overview of the available evidence from the literature.